# Dietary Intake of a Milk Sphingolipid-Rich MFGM/EV Concentrate Ameliorates Age-Related Metabolic Dysfunction

**DOI:** 10.3390/nu17152529

**Published:** 2025-07-31

**Authors:** Richard R. Sprenger, Kat F. Kiilerich, Mikael Palner, Arsênio Rodrigues Oliveira, Mikaël Croyal, Marie S. Ostenfeld, Ann Bjørnshave, Gitte M. Knudsen, Christer S. Ejsing

**Affiliations:** 1Department of Biochemistry and Molecular Biology, VILLUM Center for Bioanalytical Sciences, University of Southern Denmark, 5230 Odense, Denmark; 2Neurobiology Research Unit, Copenhagen University Hospital Rigshospitalet, 2100 Copenhagen, Denmark; 3Nantes Université, CNRS, INSERM, l’institut du thorax, 44000 Nantes, France; 4Nantes Université, CHU Nantes, Inserm, CNRS, SFR Santé, Inserm UMS 016, CNRS UMS 3556, 44000 Nantes, France; 5Arla Foods Ingredients Group P/S, 8260 Viby J, Denmark; 6Department of Clinical Medicine, Faculty of Health and Medical Sciences, University of Copenhagen, 2200 Copenhagen, Denmark; 7Cell Biology and Biophysics Unit, European Molecular Biology Laboratory, 69117 Heidelberg, Germany

**Keywords:** dietary supplementation, milk fat globule membranes, extracellular vesicles, lipidomics, ceramide biomarkers, hyperlipidemia, lipoprotein particles, lipidomic flux analysis

## Abstract

**Background/Objectives:** Nutraceuticals containing milk fat globule membranes (MFGMs) and extracellular vesicles (EVs) are purported to abate age-related metabolic dysfunction due to their richness in milk sphingolipids. As such, nutraceuticals offer a compelling strategy to improve metabolic health through dietary means, especially for elderly persons who are unable to adhere to common therapeutic interventions. To address this, we examined the effects of supplementing aged sedentary rats with an MFGM/EV-rich concentrate. **Methods**/**Results:** In a 25-week study, 89-week-old male rats received either a milk sphingolipid-rich MFGM/EV concentrate or a control supplement. Analysis of metabolic health using a battery of tests, including MS^ALL^ lipidomics of plasma, liver, and other peripheral tissues, revealed that MFGM/EV supplementation promotes accretion of unique sphingolipid signatures, ameliorates ceramide biomarkers predictive of cardiovascular death, and has a general lipid-lowering effect. At the functional level, we find that these health-promoting effects are linked to increased lipoprotein particle turnover, showcased by reduced levels of triglyceride-rich particles, as well as a metabolically healthier liver, assessed using whole-body lipidomic flux analysis. **Conclusions:** Altogether, our work unveils that MFGM/EV-containing food holds a potential for ameliorating age-related metabolic dysfunction in elderly individuals.

## 1. Introduction

The global population is projected to exceed 9 billion by 2050, largely due to rising life expectancy [1]. This demographic shift is leading to a significant rise in the elderly population, with individuals aged 60 and older expected to double to over 2 billion over the next 25 years. As the population ages, the prevalence of age-related metabolic dysfunction [2,3] and its sequelae of cardiometabolic disorders (e.g., obesity, steatohepatitis and cirrhosis, type 2 diabetes, and coronary artery disease) is also rising [4]. This inevitably places an increasing socioeconomic burden on healthcare systems worldwide. While effective pharmacological treatments and lifestyle interventions, such as physical exercise, reduce morbidity and mortality related to cardiometabolic diseases [5,6], these regimens can be difficult to implement in elderly individuals with high medication burdens, undesirable side effects of drugs, and mobility limitations. In this context, nutraceuticals, unique food constituents that have health benefits beyond their nutritional value, can serve as a compelling complementary approach to support cardiometabolic health.

As nutraceuticals, milk sphingolipids are associated with the amelioration of age-related metabolic dysfunction and improved cardiometabolic health [7,8,9]. The mechanisms underlying these effects remain largely unknown but are plausibly linked to the essential roles that these lipids play as structural components of biological membranes and signaling molecules [10,11]. Several epidemiological studies suggest that dairy foods can alleviate cardiometabolic risk factors in elderly individuals [12]. Furthermore, clinical trials have shown that intake of food rich in milk sphingolipids can lower risk factors in elderly persons predisposed to cardiovascular events [12,13]. Notably, a recent study of overweight postmenopausal women demonstrated that consumption of a milk sphingolipid-containing cream cheese reduces risk markers [14,15], including lowered plasma levels of cholesterol lipids, triglycerides, and ceramide (Cer) 18:1;2/24:1 is a clinical risk marker for cardiovascular death [16,17] (An overview of lipid class abbreviations used for lipid molecules is provided in Appendix A). These health-promoting effects were linked to a reduced intestine-derived chylomicron (CM) load following dietary intake, which in itself is a risk marker of cardiovascular disease (CVD) [18,19]. Additionally, milk sphingolipids can serve as a dietary source for the protective ceramide biomarker Cer 18:1;2/24:0, which helps reduce the risk of fatal cardiovascular outcomes [16], as well as its close structural analogs, Cer 18;1;2/23:0 and Cer 17:1;2/24:0 [20,21,22,23], which have not yet been tested for their abilities to reduce CVD risk. Despite these emerging links between the intake of milk sphingolipids and their health-promoting effects in elderly individuals, the molecular underpinnings of these benefits remain poorly understood.

A major source of milk sphingolipids is whey-based concentrates rich in milk fat globule membranes (MFGMs) and extracellular vesicles (EVs) [8,24,25]. MFGM/EVs are colloidal particles that are produced by mammary epithelial cells and contain a high proportion of lipid molecules, especially very-long-chain-containing sphingomyelins and lactosylceramides [26,27,28]. Following their digestion and uptake, these sphingolipids can serve as structural building blocks for the biosynthesis of lipids involved in a variety of physiological processes, including energy production, membrane formation, and signal transduction events [11]. This is corroborated by several intervention studies in animals and humans showing that milk sphingolipids from MFGM/EV-containing concentrates, or the structural components of these lipids, are taken up and incorporated into tissues at the whole-body level [20,21,22,23]. Moreover, we recently demonstrated that adolescent mice fed an MFGM/EV-rich diet had a metabolically healthier liver with a greater flux through metabolic pathways for the production of phosphatidylcholine lipids [20], which, in hepatocytes, are functionally coupled to the production of very-low-density lipoprotein (VLDL) and high-density lipoprotein (HDL) particles [29,30,31]. Collectively, these findings suggest that MFGM/EV supplementation may provide health benefits in the context of age-related metabolic dysfunction and related diseases in sedentary elderly individuals.

Here, we address this issue by investigating whether long-term dietary intervention with a milk sphingolipid-rich MFGM/EV concentrate improves the metabolic health of aged sedentary male rats (i.e., quiescent rats housed under regular conditions without exercise regimes). We uncovered that MFGM/EV supplementation promotes accretion of unique sphingolipid signatures at the whole-body level and that this translates to the amelioration of numerous cardiometabolic risk factors. Altogether, our findings suggest that nutraceuticals based on milk sphingolipid-rich MFGM/EV concentrates hold a therapeutic potential for ameliorating age-related metabolic dysfunction in elderly individuals.

## 2. Materials and Methods

### 2.1. Chemicals

Cow milk-derived MFGM/EV-rich whey concentrate (derived from skimmed milk) [28], Arla^®^ MIPRODAN 40 calcium caseinate, and Lacprodan^®^ SP-9224 whey protein isolate were procured from Arla Food Ingredients (Viby J, Denmark). Methanol, 2-propanol, and water were purchased from Biosolve BV (Valkenswaard, The Netherlands). Chloroform was acquired from Rathburn Chemicals (Walkerburn, UK). Ammonium formate, ^2^H_9_-choline chloride, ^13^C_3_-choline chloride, and ^13^C_6_-glucose were obtained from Sigma-Aldrich (Buchs, Switzerland). All solvents and chemicals were of the highest analytical grade. Synthetic lipid standards were purchased from Avanti Polar Lipids (Alabaster, AL, USA).

### 2.2. Preparation of Supplements

Nutritional drinks with supplements were prepared at room temperature in an airflow bench. The MFGM/EV drink was prepared as a 10% (*w*/*v*) suspension by dissolving the MFGM/EV-rich ingredient in tap water. To have an equal total protein level, the control drink was prepared as an 8.26% (*w*/*v*) suspension by dissolving equal amounts (1:1, *w*/*w*) of the whey protein isolate and the caseinate in tap water. These nutritional drinks were made on the days of utilization and poured into sterile water drinking bottles. The energy contents of the MFGM/EV-rich concentrate and whey protein–caseinate blend were 1742 and 1574 kJ/100 g, respectively (Appendix A).

### 2.3. In Vivo Study in Rats

The in vivo study was conducted in accordance with European standards and the Danish Law on Animal Experiments, approved by the Danish Animal Experiment Inspectorate (protocol 2020-15-0201-00728; date of issue: 21 December 2020), and carried out at the laboratory animal facility of the Copenhagen University Hospital (Rigshospitalet). Male Wistar rats were obtained from Janvier Labs (Le Genest-Saint-Isle, France) and allowed to acclimatize for one week with ad libitum access to water and regular chow. Moreover, the rats were maintained in pairs of two per cage and at a constant temperature (23 ± 1 °C) and humidity (40 ± 5%) under a 12 h light/12 h dark cycle. The rationale for using only male rats was due to budgetary constraints, as we minimized long-term housing costs by using commercially available aged rats, which were only offered as males.

A 25-week intervention study was carried out with elderly rats (62–63 weeks old at arrival). These rats were allocated into two dietary groups, each with 22 animals housed in 11 cages. During the first eight weeks, each pair of rats received 300 mL of the MFGM/EV drink or the control drink three times per week by giving access to an additional drinking bottle in the cage. For the remainder of this study, the pairs of rats received 200 mL of the nutritional drinks three times per week. This down-scaling was performed to avoid exhausting the limited amount of available MFGM/EV ingredient prior to the end of the intervention study. The bottles with nutritional drinks were generally emptied within 48 h. Across the study period, the MFGM/EV-supplemented and control rats received a daily dose of approximately 365 mg and 8 mg of milk phospholipids, respectively, per kg of body weight (Appendix A). The body weights of rats were measured weekly starting six weeks after onset and through to the end of this study (25 weeks). After 24 weeks, the rats were subjected to the Morris Water Maze test [32]. Nineteen (19) and eighteen (18) rats from the MFGM/EV-fed group and control group, respectively, survived throughout the study period. In the 25th and final week, 5 rats per dietary group were randomly chosen for an oral choline tracer test (OCTT) (see details below). Moreover, another 10 rats from each group were randomly chosen for the sampling of blood plasma and tissues for the analysis of steady-state lipid and metabolite levels as well as lipoprotein particles.

### 2.4. Sampling for Steady-State Lipidomics and Metabolomics

For the analyses of steady-state levels, the rats were fasted overnight (~18 h) and then euthanized. Blood plasma was collected via cardiac puncture, followed by whole-body perfusion using ~200 mL of 4 °C phosphate-buffered saline. Biopsies were then taken from the liver, duodenum, lungs, and kidneys. These samples were stored in 2 mL Eppendorf tubes, snap-frozen in dry ice, and kept at −80 °C until further processing.

### 2.5. Sampling of Blood Plasma for OCTT

For the OCTT, the rats were fasted overnight, baseline blood plasma samples were collected, and subsequently, the animals were dosed orally with 53 mg/kg ^2^H_9_-choline chloride, formulated in a suspension with 5% (*w*/*v*) MFGM/EV ingredient in sterile tap water. Afterward, the rats were provided access to food, and serial sampling of blood plasma via the jugular vein was performed after 1, 2, 5, 30, 51, and 75 h, as well as after 96 h, at the terminal timepoint by cardiac puncture. Notably, the rats were fasted for ~4.5 h prior to the plasma sampling at the 30, 51, and 75 h timepoints, and overnight until the 96 h timepoint. Whole blood was collected in K_3_EDTA-coated microvettes, and the plasma was prepared by centrifugation (3000× *g*; 15 min; 4 °C). After centrifugation, the plasma samples were collected in 2 mL Eppendorf tubes, which were snap-frozen in dry ice and stored at −80 °C until further processing.

### 2.6. Sample Preparation for Steady-State Lipidomic and Metabolomics

The blood plasma and tissue samples were subjected to two-step lipid extraction and MS^ALL^-based lipidomic analysis, as previously described [31,33,34]. The sample preparation was carried out in a cold room at 4 °C using cold solvents and consumables. The plasma samples (5 μL) were mixed with 195 μL of 155 mM ammonium formate buffer and spiked with 30 μL of an internal lipid standard mixture as well as 10 μL of an internal metabolite standard mixture with ^13^C_3_-choline and ^13^C_6_-glucose. The tissue biopsies were homogenized at 80 °C in 155 mM ammonium formate using an ULTRA-TURRAX^®^ (IKA, Staufen, Germany), as previously described [31]. Aliquots (70 μL) of the homogenates corresponding to 0.1 mg of liver, 0.1 mg of kidney, 0.25 mg of lung, and 0.35 mg of duodenum (22 μg of total protein) were mixed with 130 μL of 155 mM ammonium formate buffer and spiked with 30 μL of an internal lipid standard mixture. All samples were extracted by adding 990 μL of chloroform/methanol (15:1, *v*/*v*), followed by vigorous shaking (1400 rpm; 2 h; 4 °C) and centrifugation (1000× *g*; 2 min; 4 °C), to promote phase separation. The lower organic phase, denoted as the apolar lipid extract, was collected in a new vial. The remaining aqueous phase was reextracted with 990 μL of chloroform/methanol (2:1, *v*/*v*) for 1.5 h, followed by centrifugation to promote phase separation. The lower organic phase, denoted as the polar lipid extract, was collected in a new vial. An aliquot of the remaining aqueous phase (350 μL) was also collected in a new vial for the extracts of plasma samples. All extracts were vacuum-evaporated and stored at −20 °C until analysis.

### 2.7. Steady-State Lipidomics

The lipid extracts were dissolved in 100 μL of chloroform/methanol (1:2, *v*/*v*), gently shaken (700 rpm; 3 min), and centrifuged (13,000 rpm; 3 min). The apolar lipid extracts (10 μL) were loaded into 96-well plates and mixed with 12.9 µL of 13.3 mM ammonium formate in 2-propanol for analysis in positive-ion mode and 12.9 µL of 1.33 mM ammonium formate in 2-propanol for analysis in negative-ion mode. The polar lipid extracts (10 μL) were loaded into 96-well plates and mixed with 0.01% methylamine in methanol for analysis in negative-ion mode. The lipid extracts were analyzed by MS^ALL^ analysis in positive- and negative-ion mode using an Orbitrap Fusion Tribrid (Thermo Fisher Scientific, San Jose, CA, USA) equipped with a robotic nanoflow ion source, TriVersa NanoMate (Advion Biosciences, Ithaca, NY, USA). The apolar extracts were infused using a back pressure of 1.25 psi and an ionization voltage of ±0.95 kV. The polar extracts were infused using a back pressure of 0.6 psi and an ionization voltage of −0.96 kV. High-resolution survey Fourier-transform MS (FTMS^1^) spectra were recorded across the range of *m/z* 280 to 1400 using a max. injection time of 100 ms, automated gain control at 1 × 10^5^, three microscans, and a target resolution of 500,000. Consecutive FTMS^2^ spectra were acquired for all precursors in the range of *m/z* 398.3 to 1000.8 in steps of 1.0008 Da, recorded across a range starting from *m/z* 150 until the precursor *m/z* value of + 10 Da, using a max. injection time of 100 ms, automated gain control at 5 × 10^4^, one microscan, a target resolution of 30,000, HCD fragmentation, and a quadrupole ion isolation width of 1.0 Da. Targeted monitoring of ammoniated cholesterol (*m/z* 404.38869) and its internal standard cholesterol+^2^H_7_ (*m/z* 411.43265) was performed by MSX analysis using a max. injection time of 600 ms, automated gain control at 5 × 10^4^, five microscans, a target resolution of 120,000, HCD fragmentation at 8%, and a quadrupole ion isolation width of 1.5 Da for each precursor [34].

### 2.8. Steady-State Metabolomics

The metabolite extracts were dissolved in 100 µL of methanol/water (5:1, *v*/*v*) and gently shaken (700 rpm; 5 min) and centrifuged (20,000 rpm; 3 min). The extracts (10 μL) were loaded into 96-well plates and analyzed directly by MS^ALL^ analysis in positive-ion mode using an Orbitrap Fusion Tribrid (Thermo Fisher Scientific) equipped with a robotic nanoflow ion source, TriVersa NanoMate (Advion Biosciences). The extracts were infused using a back pressure of 0.75 psi and an ionization voltage of 1.0 kV. High-resolution survey Fourier-transform MS (FTMS^1^) spectra were recorded across the range of *m/z* 50 to 450 using a max. injection time of 100 ms, automated gain control at 1 × 10^5^, three microscans, and a target resolution of 500,000. Consecutive FTMS^2^ spectra were acquired for all precursors in the range of *m/z* 60 to 424 in 1 Da steps, using a max. injection time of 100 ms, automated gain control at 5 × 10^4^, one microscan, a target resolution of 120,000, HCD fragmentation, and a quadrupole ion isolation width of ±0.5 Da.

### 2.9. Lipidomic Flux Analysis of OCTT Samples

Plasma samples (5 μL) from the OCTT were extracted and analyzed essentially as described above. Notably, a higher target resolution of 240,000 was used for the FTMS^2^ analysis. Otherwise, instrument settings were similar to those specified in the previous section.

### 2.10. Profiling of Lipoprotein Particles

The plasma samples were analyzed by gel permeation chromatography coupled to online monitoring of total levels of glyceride and sterol levels at Immuno-Biological Laboratories (Gunma, Japan), as previously described [35,36].

### 2.11. Lipid Quantification

The identification and quantification of lipid molecules was performed using the ALEX123 software and a data-processing pipeline in SAS 9.4 (SAS Institute, Cary, NC, USA) [33,37,38,39]. Briefly, lipid molecules detected by full-scan FTMS were identified using a maximum *m/z* tolerance of ±0.0040 amu, corrected for potential ^13^C isotope interference, and reported at the “species level”. Lipid molecules were confirmed through molecular lipid species-specific fragments and lipid class-specific fragments detected by FTMS^2^ analysis using a maximum *m/z* tolerance of ±0.0065 amu [38,39]. For the analysis of OCTT data, a maximum *m/z* tolerance of ±0.0015 amu was used for the identification of stable isotope-labeled phosphocholine fragments derived from labeled choline-containing lipid precursor ions (i.e., *m/z* 193.1298 for lipids labeled with ^2^H_9_, *m/z* 190.1110 for lipids labeled with ^2^H_6_, *m/z* 187.0922 for lipids labeled with ^2^H_3_, and *m/z* 184.0733 for unlabeled lipids). Identified lipid molecules were quantified by normalizing their intensities to those of the respective internal lipid standards, subsequent multiplication by the amounts of the respective lipid standards, and normalization to the extracted sample amount (i.e., µL of plasma, mg of tissue, or µg of protein).

### 2.12. Metabolite Quantification

The quantitative profiling of metabolites was carried out using the ALEX123 software with a manually curated metabolite database and a data-processing pipeline in SAS 9.4 (SAS Institute). Sodiated glucose and the internal ^13^C_6_-glucose standard were monitored by high-resolution FTMS^1^ analysis. Choline, protonated betaine, protonated dimethylglycine, and a ^13^C_3_-labeled choline internal standard were monitored by high-resolution FTMS^2^ analysis. The absolute level of glucose was quantified by normalizing its intensity to that of the internal ^13^C_6_-glucose standard, multiplication by the molar spike amount (10 nmol), and, finally, normalization to the extracted amount of plasma (5 μL). The absolute levels of the other metabolites were estimated by normalizing the sum of their fragment intensities to the sum of the fragment intensity of the internal ^13^C_3_-choline standard, multiplication by the molar spike amount (100 pmol), and normalization to the extracted amount of plasma.

### 2.13. Proteomics

Untargeted analysis of the plasma proteome and a targeted quantitative analysis of ApoB48 and ApoB100 were performed using plasma samples. For targeted analysis, a mixture of synthetic unlabeled peptides (ApoB48: LSQLETYAI; ApoB100: ASEAVYDYVK (Thermo Fisher Scientific)) was prepared and serially diluted to generate seven standard solutions ranging from 0.5 µmol/L to 10 µmol/L. The isotopically labeled peptides LSQLETYA-[^13^C_6_^15^N]I and ASEAVYDYV-[^13^C_6_^15^N_2_]K were used as internal standards and added to the digestion buffer prior to the sample preparation. The standard solutions and plasma samples were processed using the ProteinWorks™ eXpress kit (Waters Corporation, Milford, MA, USA), as described previously [40,41]. Briefly, 20 µL of each sample was reduced, alkylated, and digested overnight with trypsin. The resulting peptides were purified in 30 mg Oasis HLB cartridges (Waters Corporation), dried under nitrogen, and reconstituted in 100 µL of 5% acetonitrile containing 0.1% formic acid.

Targeted analyses were performed using an Xevo TQ Absolute triple quadrupole mass spectrometer (Waters Corporation) equipped with an electrospray ionization interface and coupled to an Acquity Premier UPLC system (Waters Corporation). Peptide separation was achieved in an Acquity Premier BEH C_18_ column (2.1 × 100 mm, 1.7 µm, VanGuard™ FIT; Waters Corporation), maintained at 60 °C, using a linear gradient of mobile phase B (100% acetonitrile with 0.1% formic acid) in mobile phase A (5% acetonitrile with 0.1% formic acid), as previously described. Peptides were detected in positive-ion mode under the following conditions: a capillary voltage of 3 kV; a desolvation gas (N_2_) flow rate of 650 L/h; a desolvation temperature of 450 °C; and a source temperature of 120 °C. The multiple reaction monitoring mode was used for detection (Appendix A). Data acquisition and analysis were performed using the MassLynx and TargetLynx software (version 4.1; Waters Corporation). Quantification was based on the chromatographic peak area ratios of unlabeled to labeled peptides, and calibration curves were established from the standard solutions.

Untargeted proteomic analyses were carried out on the same plasma digests using label-free quantification by nano-LC-MS/MS [41]. The samples were analyzed with an Exploris 480 mass spectrometer (Thermo Fisher Scientific) equipped with a nanoelectrospray ion source and coupled to a Vanquish Neo UHPLC system (Thermo Fisher Scientific). Peptides were separated in a 75 μm × 500 mm C_18_ column (EASY-Spray PepMap; Thermo Fisher Scientific). Data acquisition was performed using the Xcalibur software (version 2.9; Thermo Fisher Scientific). Protein identification was realized using Proteome Discoverer (version 2.5; Thermo Fisher Scientific) through a database search against the rat proteome. Trypsin/P was selected as the proteolytic enzyme, allowing up to two missed cleavages and a peptide length of greater than six amino acids. The precursor and fragment mass tolerances were set to ±10 ppm and ±20 ppm, respectively. The following peptide modifications were considered during the search: carbamidomethylation of cysteine (+57.0214 Da, fixed), *N*-terminal acetylation (+42.0105 Da, variable), and methionine oxidation (+15.9949 Da, variable).

### 2.14. Statistical Analysis

Statistical analyses were carried out using SAS 9.4 (SAS Institute). This included ANOVA with post hoc testing using multiple hypothesis correction (PROC GLM), non-parametric testing (PROC NPAR1WAY), and mixed-effects modeling (PROC MIXED). A *p*- or q-value of less than 0.05 was considered statistically significant. Outliers were identified and removed using the interquartile range test prior to statistical testing. Visualization of the data was realized using Tableau Desktop 2024 (Tableau Software) or SAS 9.4.

## 3. Results and Discussion

### 3.1. Study Design

To explore the metabolic effects of supplementing aged rats with the milk sphingolipid-rich MFGM/EV-based concentrate, we carried out a 25-week intervention study (Figure 1A). Here, elderly male rats, 63–64 weeks (~14.6 months) old, at the beginning of this study, were allocated into a treatment group or a control group. Rats in the treatment group were supplemented three times per week with a nutritional drink containing the MFGM/EV concentrate. Rats in the control group were supplemented three times per week with a drink containing a milk lipid-deficient whey protein–caseinate blend. Throughout this study, the sedentary rats (i.e., regularly housed rats not subjected to exercise regimens) were also fed a base diet ad libitum composed of regular chow. This intervention study included longitudinal monitoring of body weight (Figure 1A). At the end of this study, we carried out an oral choline tracer test (OCTT), together with lipidomic flux analysis, to assess the metabolic fitness of the rats. We also performed cognitive tests, involving physical activity, the results of which are reported separately [32]. Following euthanasia, samples of multiple tissues were collected for lipidomic analysis, as well as blood plasma for profiling metabolites and lipoprotein particles.

Over the course of this study, body weight demonstrated no significant difference between the two dietary regimes (Figure 1B). However, physical activity from the cognitive testing resulted in significant weight loss for both groups, which was marginally greater for the MFGM/EV-fed rats compared with the control rats (Figure 1C). Testing the fasting blood glucose level, a proxy for glucose intolerance, showed no difference (Figure 1D). Since the MFGM/EV concentrate is a source of choline [28], we also assessed the plasma levels of choline and its metabolites, betaine and dimethylglycine. The levels of these were also not different between the two groups (Figure 1D). Altogether, these data show that the MFGM/EV supplementation was well-tolerated by the aged rats and suggest that the concentrate might have health-promoting effects, given the slightly greater weight loss in response to physical activity in the treatment group.

### 3.2. MFGM/EV Supplementation Promotes Accretion of Milk-Related Sphingolipids in Blood Plasma

Next, we examined whether the dietary intervention elevated levels of very-long odd-chain sphingolipids, a change we previously observed in adolescent mice [20]. To this end, we carried out in-depth MS^ALL^ lipidomic analysis [33] of blood plasma collected after 25 weeks of supplementation. This analysis quantified (i.e., pmol/μL plasma) 673 lipid molecules, encompassing 21 lipid classes (Appendix A). A total of 417 lipid molecules were monitored at the molecular species level, with assignments of individual hydrocarbon chains (e.g., PC 16:0–18:2), and 256 lipids were monitored at the species level with assignment of the total number of C atoms, double bonds, and hydroxyl groups in all hydrocarbon chains (e.g., PE 38:4).

To survey overall effects, we carried out an analysis of variance (ANOVA) with multiple hypothesis correction. Using a stringent statistical threshold with a q-value of <0.01 (i.e., a multiple hypothesis-corrected *p*-value), we shortlisted a total of 96 lipids with significantly altered abundance (Appendix A). Using these lipid concentrations for unsupervised principal component analysis demonstrated that the plasma lipid profiles of the MFGM/EV-supplemented rats were clustered together and separately from the control rats (Figure 2A).

To examine the metabolic differences in further detail, we generated a volcano plot, which revealed that the lipid profiles of the MFGM/EV-fed rats were unique in having primarily elevated levels of sphingolipids and glycerophospholipids, whereas the lipids elevated in the aged control rats belonged to all lipid categories (Figure 2B and Appendix A). More specifically, for the MFGM/EV-fed rats, we found that the elevated lipids were primarily sphingomyelins, ceramides, and phosphatidylcholines, with odd-numbered sphingoid and fatty acyl chains (e.g., Cer 17:1;2/24:1, SM 18:1;2/23:0, and PC 15:0–18:2), which was similar to our observation for the MFGM/EV-fed mice. In contrast, the control rats had higher levels of triglycerides, cholesteryl esters, phosphatidylcholines, sphingomyelins, and ceramides, with even-numbered fatty acyl and sphingoid chains (e.g., TAG 62:11, CE 22:4, PC 16:0–20:4, SM 18:1;2/24:1, and Cer 18:1;2/24:1) (Appendix A).

To explore the lipid metabolic effects in further detail, we inspected the overlap between elevated lipids in the aged rats and adolescent mice fed the MFGM/EV-rich concentrate [20]. Notably, 37 of 86 lipids consistently elevated in mouse plasma were also increased in the plasma of the aged MFGM/EV-fed rats (Figure 2C). Among these were 12 sphingolipids (e.g., 5 ceramides and 6 sphingomyelins), 18 glycerophospholipids (e.g., 11 phosphatidylcholines), and 5 cholesteryl esters (i.e., sterol lipids) (Figure 2D). Notably, these lipid molecules featured primarily odd-numbered fatty acyl and sphingoid chains (Figure 2E).

Strikingly, among the common lipids, we again found several very-long-odd-chain sphingolipids, including Cer 18:1;2/23:0 and its isomer Cer 17:1;2/24:0, as well as the downstream metabolic products SM 18:1;2/23:0 and HexCer 18:1;2/23:0 (Figure 2E). Furthermore, we also found Cer 18:1;2/23:1 and its isomer Cer 17:1;2/24:1, as well as the metabolic product SM 18:1;2/23:1. The concentrations of these sphingolipids were generally low in the control rats, ranging from ~0.5 pmol/μL for the ceramides to ~6 pmol/μL for the sphingomyelins (Figure 2F). The intake of the MFGM/EV concentrate increased these sphingolipids by 2.4- to 8.7-fold, which is similar to what we observed in mice. Overall, these data show that the MFGM/EV supplementation increased the plasma levels of sphingolipids with molecular structures identical to those of prominent sphingolipids in the MFGM/EV concentrate (e.g., 18:1;2/23:0 and 17:1;2/24:0) [28]. Additionally, the similarities between the aged rats and adolescent mice demonstrate that the effects of the MFGM/EV supplementation are generalizable.

### 3.3. Accretion of Unique Sphingolipid Signatures in Tissues

Our study with mice also revealed that an MFGM/EV-rich diet prompts a variety of lipid metabolic effects in tissues with different metabolic and physiological functions. To examine whether this is also the case for the aged rats, we analyzed the lipidomes of the duodenum, liver, lungs, and kidneys. On average, the lipidomic analyses quantified (e.g., pmol/mg of tissue) 661 lipid molecules covering 26 lipid classes (Appendix A). To pinpoint lipidomic differences, we carried out ANOVA with multiple hypothesis correction. A stringent statistical threshold with a q-value of <0.01 shortlisted 126, 129, 90, and 79 significantly altered lipid abundances in the duodenum, liver, lungs, and kidneys, respectively (Figure 3A). Altogether, this shows that the MFGM/EV supplementation affected the lipid metabolic activities of primarily the duodenum and liver and, to a lesser degree, those of the lungs and kidneys (Figure 3A–E).

To better understand the metabolic effects of the MFGM/EV concentrate at the systemic level, we examined the overlap between lipid molecules that increased concurrently in blood plasma and the four tissues. To do so, we selected lipids that were significantly elevated in the MFGM/EV treatment group (q-value < 0.05 and fold change > 1) and shortlisted lipids that were elevated in four or five of the biopsies. This list featured a total of 27 lipid molecules, whereof 26 were increased in the duodenum, 16 were increased in the plasma, 23 were increased in the liver, 25 were increased in the lungs, and 24 were increased in the kidneys (Figure 3F and Appendix A). Among the common lipids, 10 were sphingolipids (e.g., 4 ceramides and 5 sphingomyelins) and 16 were glycerophospholipids (e.g., 10 phosphatidylcholines). Remarkably, and similar to our finding for mice, these molecules featured primarily odd-numbered fatty acyl and sphingoid chains (e.g., Cer 17:1;2/24:0 and PE 17:0–20:4).

Upon closer inspection of the lipid molecules, we again observed a striking molecular relationship between the subset of odd-chain sphingolipids. Across all tissues, we consistently identified Cer 18:1;2/23:0 and its downstream metabolic products SM 18:1;2/23:0 and HexCer 18:1;2/23:0 (Figure 3F). Similarly, Cer 18:1;2/23:1 and its metabolic product SM 18:1;2/23:1 were also consistently elevated in MFGM/EV-fed rats. The concentrations of these sphingolipids were generally low in the tissues of the aged control rats (Figure 4). The intake of the MFGM/EV supplement consistently increased these sphingolipids by 1.7- to 5.9-fold. Notably, several other odd-chain ceramide and sphingomyelin species were also elevated across all the tissues (Figure 3F; Appendix A). Taken together, these observations mirror our previous finding in mice and underpin a causal relationship between prominent very-long-odd-chain sphingolipid constituents in the ingested MFGM/EV-rich concentrate [28] and the specific accretion of endogenous sphingolipids having identical backbone structures (e.g., 18:1;2/23:0 and 17:1;2/24:0).

### 3.4. High-Risk CVD Biomarkers Are Reduced by MFGM/EV Supplementation

Certain plasma ceramides are high-risk biomarkers and purported mediators of cardiometabolic morbidity and mortality in humans (Figure 5A) [16,17,42]. Given that MFGM/EV supplementation influences levels of ceramides at the whole-body level, we assessed whether the dietary intervention also modulates plasma ceramide biomarkers of CVD in aged rats. Notably, the molar ratio between the even-numbered long-chain Cer 18:1;2/16:0 species and the very-long-chain Cer 18:1;2/24:0 species is a high-risk marker predictive of a fatal cardiovascular outcome. Likewise, the molar ratio between Cer 18:1;2/24:1 and Cer 18:1;2/24:0 is also a high-risk marker. Assessing these high-risk markers demonstrated that the MFGM/EV-fed rats had a significant 1.4-fold (−29%) lower Cer 18:1;2/16:0-based risk score (*p* = 0.038) as well as a significant 1.8-fold (−44%) lower Cer 18:1;2/24:1-based risk score (*p* = 0.00033) (Figure 5B). Underpinning this is the lowering of plasma concentrations for the risk markers Cer 18:1;2/16:0 and Cer 18:1;2/24:1, together with a constant level of the protective marker Cer 18:1;2/24:0 (Appendix A).

Importantly, MFGM/EV supplementation specifically increases levels of ceramides that are close structural analogs of the even-chain risk markers. Specifically, Cer 18:1;2/23:0 and Cer 17:1;2/24:0 are potential protective markers as they closely mimic Cer 18:1;2/24:0 by a single methylene unit in their hydrocarbon chains (Figure 5C). Similarly, Cer 18:1;2/15:0 is a potential risk marker as it resembles Cer 18:1;2/16:0, and Cer 18:1;2/23:1 and Cer 18:1;2/24:1 are potential risk markers as they mimic Cer 18:1;2/24:1.

To determine whether the reduction in the original CVD risk scores (Figure 5B) is biased by the accretion of odd-chain ceramides from the MFGM/EV concentrate, we recalculated the high-risk scores by factoring in the plasma concentrations of the different odd-chain analogs. Notably, this revealed a more pronounced and significant 1.6-fold (−38%) lowering (*p* = 0.0022) of the long-chain risk score that considers 16:0- and 15:0-ceramides relative to 24:0- and 23:0-ceramides (Figure 5D). Concordantly, the very-long-chain risk score, considering 24:1- and 23:1-ceramides relative to 24:0- and 23:0-ceramides, also showed a more pronounced and significant 1.9-fold (−47%) lowering (*p* = 0.00018) (Figure 5D). This lowering of the high-risk scores is primarily driven by the increase in the plasma concentration of the protective analog Cer 18:1;2/23:0 (Appendix A). Altogether, these data show that the intake of the MFGM/EV-rich concentrate ameliorates high-risk markers of CVD death in aged sedentary rats. Furthermore, these data indicate that the diagnostic utility of the high-risk markers, which currently considers only even-chain ceramides [42], should be extended to also monitor levels of the odd-chain ceramide analogs, to allow better stratification of individuals who frequently consume dairy products [43].

### 3.5. MFGM/EV Supplementation Improves Biomarkers of Cardiometabolic Health

To assess whether the lowering of the ceramide biomarkers translates to improvement of other, more conventional lipid biomarkers, we examined whether the dietary supplementation also modulates the concentration of all lipids as well as the total levels of glycerides (i.e., diglycerides and triglycerides), non-esterified fatty acids, and cholesterol lipids in plasma and each of the tissues. Overall, this revealed that the MFGM/EV concentrate had a general lipid-lowering effect (Appendix A). For example, the supplementation led to a 1.3-fold (*p* = 0.070) and 1.1-fold (*p* = 0.070) lower total lipid concentration in the plasma and liver, respectively (Figure 5E,F). In the plasma, this effect was primarily caused by a significant reduction in the cholesteryl ester level (*p* = 0.049) (Figure 5G) and a marginal lowering of the non-esterified fatty acid level (*p* = 0.082) (Figure 5H). In the liver, the effect was primarily due to a subtle lowering of the total glyceride level (*p* = 0.058) (Figure 5I), which was compounded by a significant reduction in the diglyceride level (*p* = 0.0013) (Figure 5J) and a marginal lowering of the triglyceride level (*p* = 0.070) (Figure 5K). Further supporting the notion of a lipid-lowering effect, we also found lower lipid levels in the other tissues, including a significant reduction in the total glyceride level in the duodenum (5.5-fold; *p* = 0.014) and lungs (6.3-fold; *p* = 0.019) as well as a marginal reduction in the kidneys (1.4-fold; *p* = 0.14) in the MFGM/EV-supplemented rats (Figure 5L–N and Appendix A).

Given that lipid markers of cardiovascular disease risk often reflect changes in lipoprotein metabolism, we examined the profile of lipoprotein particles in the two groups of aged rats. To this end, plasma samples were analyzed by gel permeation chromatography with dual monitoring of total levels of glycerides as well as cholesterol lipids (i.e., the sum of free cholesterol and cholesteryl esters) (Figure 6A) [35]. Contrary to traditional lipoprotein-selective assay kits that yield a single concentration value (e.g., LDL-C), the chromatographic approach provides a much broader overview of the full continuum of lipoprotein particles, ranging from large, intestinal-derived CMs to hepatocyte-derived VLDL particles and CM remnants, followed by smaller low-density lipoprotein (LDL) and HDL particles. The analysis demonstrated significantly different profiles for the two groups of rats. Specifically, both the glyceride- and cholesterol-based profiles were elevated in the control rats (Figure 6A), corroborating our observation that the MFGM/EV supplement has a lipid-lowering effect. More specifically, the MFGM/EV-fed rats had significantly lower glyceride levels across most groups of lipoprotein particles as well as a significantly lower level of sterol-containing HDL particles (Figure 6B).

Given the functional relationship between lipoprotein particles and the proteins that decorate their surfaces, we explored whether the observed differences in lipoprotein profiles were reflected in the apolipoprotein composition of plasma. Untargeted proteomic analysis identified twelve apolipoproteins, among which three were significantly reduced in the MFGM/EV-fed rats (Figure 6C). These proteins comprise ApoE and ApoC4, which modulate the production and clearance of triglyceride-rich lipoprotein particles (e.g., chylomicrons and VLDL) [44,45], and ApoA2, which is involved in cholesterol efflux from cells to HDL particles [46].

Surprised to find that the level of ApoB, a hallmark of chylomicrons and VLDL particles, did not differ between the two groups of rats, we carried out a targeted proteomic analysis to quantify the molar abundances of the two isoforms ApoB48 and ApoB100, which, in humans, are markers of chylomicrons and VLDL particles, respectively. This analysis showed that the molar ratio of ApoB48 to ApoB100 was 54:46 in aged control rats and 50:50 in MFGM/EV-fed rats (Figure 6D). However, this slight shift in the ApoB48-to-ApoB100 ratio is not statistically significant. Notably, in contrast with humans, the *APOB* transcript in rats is also edited in the liver, resulting in the secretion of both ApoB100- and ApoB48-containing VLDL particles [47]. Accordingly, the lack of a significant difference in the isoform ratio, intended here as a proxy for intestine- versus liver-derived triglyceride-rich lipoproteins, may be masked by the species-specific mRNA editing.

Taken together, these results suggest that supplementing aged rats with food rich in MFGM/EVs primarily lowers lipids and improves a wide range of lipid and lipoprotein biomarkers associated with cardiometabolic health, while the effects on the apolipoprotein profile are comparatively modest.

### 3.6. Intake of MFGM/EV Concentrate Increases Lipid Metabolic Turnover

Lipoprotein metabolism is tightly coupled to the dynamics of membrane lipid synthesis, especially of choline-containing lipids that decorate the surface of lipoprotein particles. Genetic ablation of the phosphatidylethanolamine *N*-methyltransferase (PEMT) pathway reduces hepatic VLDL secretion and renders mice susceptible to steatosis and liver failure [29,48]. Based on this pathophysiological association, we assessed whether intake of the MFGM/EV supplement modulates lipid metabolic turnover at the whole-body level. To do so, we subjected the aged rats to an OCTT (oral choline tolerance test) at the end of the 25 weeks of intervention. In combination with high-resolution lipidomic flux analysis, this test determines the kinetics of ^2^H_9_-choline tracer incorporation into phosphatidylcholine, lysophosphatidylcholine, and sphingomyelin molecules, which is a proxy for the flux through the CDP-choline pathway (Figure 7A). In addition, it also determines the kinetics of ^2^H_3_- and ^2^H_6_-choline incorporation, which is a proxy for the flux through the hepatic PEMT pathway [49]. Altogether, the analysis quantified 186 labeled lipids (Appendix A).

The total level of tracer incorporation showed kinetic timelines with apices in the proportion of ^2^H_9_- and ^2^H_6_-labeled lipids after 5 h, and in the proportion of ^2^H_3_-labeled lipids after 24 h (Figure 7B). Comparing the timelines revealed that the MFGM/EV-fed rats had higher production and turnover rates of all labeled lipids (Figure 7B), which indicates that intake of the MFGM/EV concentrate increases lipid metabolic flux through both the CDP-choline pathway as well as the hepatic PEMT pathway.

Next, we examined how the differences in metabolic activity affect the timelines of individual lipids. Statistical analysis showed that the timelines of 43 labeled lipids were significantly different between the two groups of aged rats (Figure 7C). The majority of these lipids were phosphatidylcholines (18), followed by sphingomyelins (14) and lysophosphatidylcholines (11). These timelines were predominantly related to lipids labeled with ^2^H_3_, which result from PEMT activity (Figure 7D). Interestingly, we found that timelines of several phosphatidylcholines with polyunsaturated acyl chains, including the archetypal PEMT product PC(+^2^H_3_) 18:0–22:6 exhibited large differences (Figure 7E). Concordantly, the timelines for the other archetypal PEMT products, PC(+^2^H_3_) 16:0–22:6 and LPC(+^2^H_3_) 22:6, were marginally elevated (Appendix A). Interestingly, the timelines of analogous PC molecules labeled with ^2^H_9_, and thus produced by the CDP-choline pathway and fatty acyl chain remodeling, were also significantly increased for the MFGM/EV-supplemented rats (Appendix A). Taken together, these findings show that intake of the MFGM/EV concentrate increases the metabolic turnover of phosphatidylcholine molecules with a 22:6 chain through both the PEMT pathway as well as the CDP-choline pathway.

Sphingomyelin molecules, such as ^2^H_3_- and ^2^H_9_-labeled analogs of the odd-numbered very-long-chain species SM 41:1;2 and SM 41:2;2, also exhibited significant differences (Figure 7F and Appendix A). This finding is consistent with the observation that MFGM/EV-fed rats have higher steady-state levels of the two 41:1;2 isomers SM 18:1;2/23:0 and SM 17:1;2/24:0, as well as the two 41:2;2 isomers SM 18:1;2/23:1 and SM 17:1;2/24:1 (Figure 2F and Appendix A). Furthermore, the timelines for ^2^H_3_- and ^2^H_9_-labeled SM 42:1;2, SM 42:2;2 and SM 34:1;2, corresponding to analogs with 18:1;2/24:0, 18:1;2/24:1, and 18:1;2/16:0 backbones, respectively, were all greater in MFGM/EV-supplemented rats compared with the control rats (Figure 7G and Appendix A). Of note, the lipidomic flux analysis was not able to identify the sphingoid and fatty acyl chains of the labeled sphingomyelin species due to their low abundance. Thus, the proportions of the labeled sphingomyelins reflect the contribution from the pool of the underlying isomers. The elevated metabolic turnover of sphingomyelins with odd- and even-numbered chains likely reflects the backbone structure of sphingolipids present in the MFGM/EV-rich supplement [28].

Altogether, these data demonstrate that intake of the MFGM/EV concentrate increases lipid metabolic activity at various levels. This includes increased flux through pathways involving PEMT that produce polyunsaturated phosphatidylcholine molecules (e.g., PC 18:0–22:6), which are used by hepatocytes to decorate the surface of secreted VLDL particles. Altogether, this suggests that the aged MFGM/EV-fed rats have metabolically healthier livers than the aged control rats, similar to what we previously observed in mice [20].

## 4. Conclusions

Age-related metabolic dysfunction increases the risk of cardiometabolic disorders, such as obesity, steatohepatitis, type 2 diabetes, and coronary artery disease, and represents an increasing health burden for elderly individuals and healthcare systems worldwide. Here, we investigated whether dietary supplementation with a milk sphingolipid-rich MFGM/EV concentrate can ameliorate age-related metabolic decline using aged sedentary male rats as a preclinical model (i.e., quiescent rats housed under regular conditions and without additional exercise regimes). Overall, our study unequivocally demonstrates that MFGM/EV supplementation leads to accretion of very-long-odd-chain ceramide and sphingomyelin molecules that are structurally identical to sphingolipid constituents in the concentrate [28]. Moreover, we showed that this lowers clinically relevant ceramide biomarkers, as well as more common lipid markers of cardiometabolic disease risk. Finally, our study reveals that the health-promoting effects are coupled to improved lipoprotein metabolism and liver function in the aged MFGM/EV-supplemented rats. Altogether, our study contributes to the emerging paradigm that the intake of MFGM/EV-containing food can lead to the build-up of bioactive very-long-chain-containing sphingolipids that have beneficial effects on whole-body metabolism in elderly individuals.

The molecular underpinnings of how milk sphingolipids impinge on metabolic health are multifaceted and poorly understood. Several intervention studies in humans, pigs, and rodents, and the present study, consistently show that intake of MFGM/EV-containing food results in accretion of very-long-odd-chain sphingolipids, as revealed by lipidomic analysis of steady-state lipid levels [20,21,22,23,50]. This, in turn, points to intestinal digestion and uptake as being a key determinant for the build-up of the unique milk sphingolipid signatures. While the digestion and uptake of long-chain (e.g., 16:0) sphingolipids have been studied [51,52,53], it is unclear how the digestive pathways handle the uptake and systemic incorporation of very-long-chain sphingolipids. A major pathway purportedly involves complete hydrolysis to the main breakdown products, a sphingoid base and a very-long-chain fatty acid, which are absorbed and can be reassembled by the endogenous metabolic machinery to produce identical sphingolipid molecules. Moreover, an alternative pathway that facilitates the direct uptake of intact very-long-chain sphingolipids and/or their partially hydrolyzed ceramide intermediates also seems to be at play. Although this remains to be tested in future studies, there are indications in the published literature that this process is taking place. For example, a study on mice with disrupted intestinal sphingomyelinase showed that a substantial proportion of a sphingomyelin-based radiotracer still accumulates in the circulation after 1 h [51]. In addition, a study with lymph-cannulated rats showed that ingestion of milk sphingomyelin leads to the build-up of ceramide intermediates with identical backbone structures in chylomicrons [54]. Intriguingly, this study also highlighted a preferential uptake of very-long-chain ceramides, including the 18:1;2/23:0 signature that is consistently found to be elevated in animals and humans fed MFGM/EV-containing food (Figure 3F) [20,21,22,23]. Altogether, this underscores the need for additional studies to dissect the molecular origins of the sphingolipid signatures that accumulate following MFGM/EV consumption.

Further work is also needed to deepen our understanding of the mechanisms that govern the health-promoting effects of milk sphingolipids [55]. One commonly suggested mechanism is that sphingolipids associate with cholesterol in the intestinal lumen and inhibit its uptake, which reduces the amount of cholesterol lipids and leads to a hypocholesterolemic effect. While our study on sedentary aged male rats supports the previous findings on the hypocholesterolemic effect of milk sphingolipids, it also sheds light on the mechanisms underlying their additional hypotriglyceridemic effect [14]. Specifically, our more detailed lipoprotein profiling data, together with the analysis of whole-body lipid metabolic flux, suggest that increased turnover of triglyceride-rich lipoprotein particles, accompanied by higher clearance as well as hepatic and intestinal production, is also at play. Specifically, our data show that aged rats have elevated CM levels during fasting and that this is lowered by MFGM/EV supplementation (Figure 6A). Furthermore, the supplementation also reduces the level of triglyceride-rich lipoprotein particles that reflect liver-derived VLDL and CM remnants. In agreement with this, the elevated lipid metabolic flux, specifically through the hepatic PEMT pathway that generates phosphorylcholine lipids for the production of VLDL particles [29], suggests a higher rate of VLDL production. How milk-derived very-long-chain sphingolipids modulate lipoprotein metabolism at the whole-body level to maintain a lower steady-state level of triglyceride-rich particles is currently unclear and warrants further mechanistic investigations. Furthermore, while this study was limited to aged male rats, potential gender-specific effects also deserve attention in future studies. Notably, however, our findings concur with results from a clinical intervention study in overweight postmenopausal women [14,15], raising the possibility that a conserved pathophysiological mechanism is at play across gender and species.

In summary, our study demonstrates that dietary supplementation with an MFGM/EV-rich concentrate is a viable strategy for modulating levels of bioactive very-long-chain ceramide biomarkers at the whole-body level. We further show that MFGM/EV supplementation can be used to ameliorate cardiometabolic risk markers in aged sedentary male rats with metabolic dysfunction. Furthermore, our findings define a framework for further exploring the molecular underpinnings of the health-promoting effects of milk sphingolipids rich in MFGM/EV-based food, as well as improving the therapeutic utility of nutraceuticals to treat cardiometabolic disorders in elderly individuals.

## Figures and Tables

**Figure 1 nutrients-17-02529-f001:**
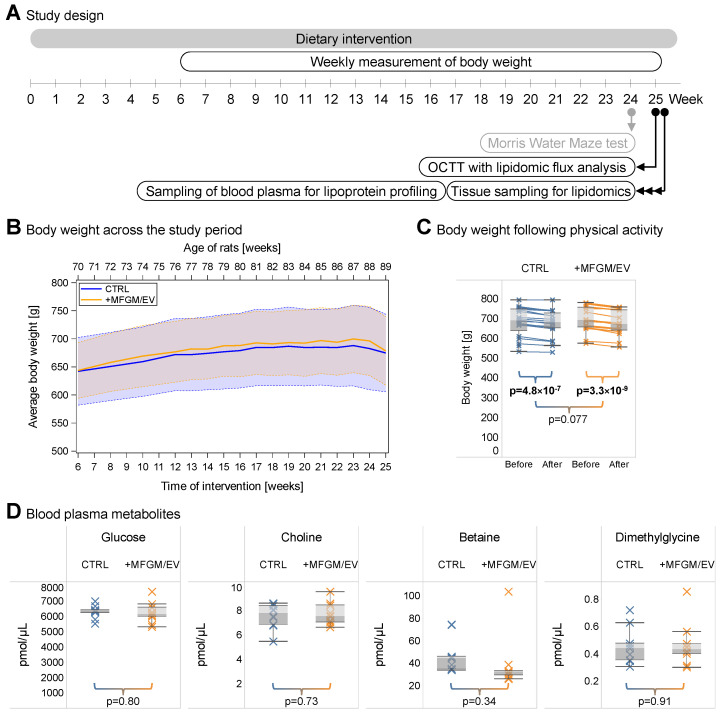
The study design and basic endpoints. (**A**) The design of the intervention study with highlights of timepoints for longitudinal and endpoint measurements. A Morris Water Maze test was carried out after 24 weeks to assess the cognitive performance of the animals. The results are reported [32]. OCTT, oral choline tracer test. (**B**) Body weight as a function of intervention time (lower *x*-axis) and rat age (upper *x*-axis). Each line plot represents the mean ± SD indicated by the band of the same color (*n* = 19–21 rats/group). A repeated-measures ANOVA F-test shows no significant differences in body weight for the two groups of rats (*p* = 0.8). (**C**) Changes in body weight following physical activity. The data are represented as box plots with medians, and crosses show individual values (*n* = 19–20 rats/group). Statistical analysis was carried out by Student’s *t*-test. Significant differences are highlighted in bold. (**D**) Levels of metabolites in plasma. The data are represented as box plots with medians, and crosses show individual values (*n* = 10 rats/group). Statistical analysis was performed by Student’s *t*-test.

**Figure 2 nutrients-17-02529-f002:**
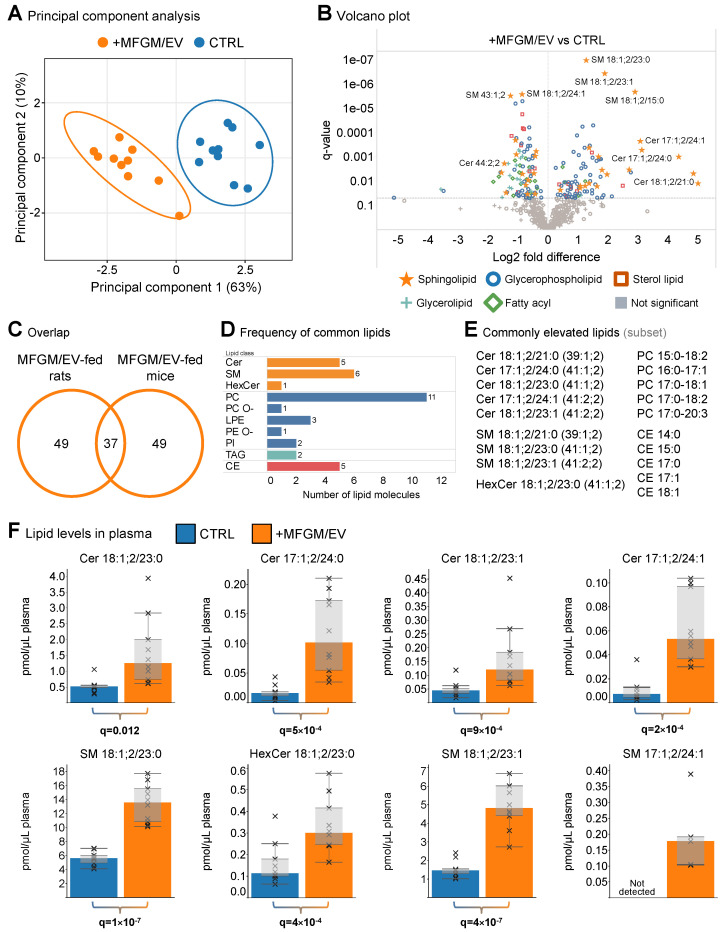
MFGM/EV supplementation elevates plasma levels of very-long-odd-chain sphingolipids. (**A**) Principal component analysis of significantly different plasma lipids (*n* = 10 rats/group). Ellipses indicate 95% confidence intervals. (**B**) Volcano plot showing lipids with significantly elevated and lowered plasma levels. The horizontal line indicates the significance level at q-value = 0.05. Statistical analysis was carried out using ANOVA with multiple hypothesis correction. (**C**) Venn diagram showing numbers of lipid molecules that are significantly elevated in plasma of MFGM/EV-fed rats and mice. The shortlist of lipids in mice is taken from [20]. The shortlists are based on q-value < 0.05 and fold change > 1. (**D**) Distribution of significantly elevated lipids that are common for MFGM/EV-fed rats and mice. (**E**) Examples of lipid molecules that are consistently elevated in both MFGM/EV-fed rats and mice. Sphingolipid molecules are denoted at both the molecular species level and species level (in parentheses) to emphasize isomeric and metabolic relationships between the ceramide (Cer), sphingomyelin (SM), and hexosylceramide (HexCer) species. (**F**) Concentrations of representative sphingolipids that are significantly and consistently elevated in plasma of rats and mice fed the MFGM/EV ingredient. Data are represented by medians, box plots, and crosses show individual values (*n* = 5–10 rats/group). Statistical analysis was performed by ANOVA with multiple hypothesis correction. Significant differences are highlighted in bold.

**Figure 3 nutrients-17-02529-f003:**
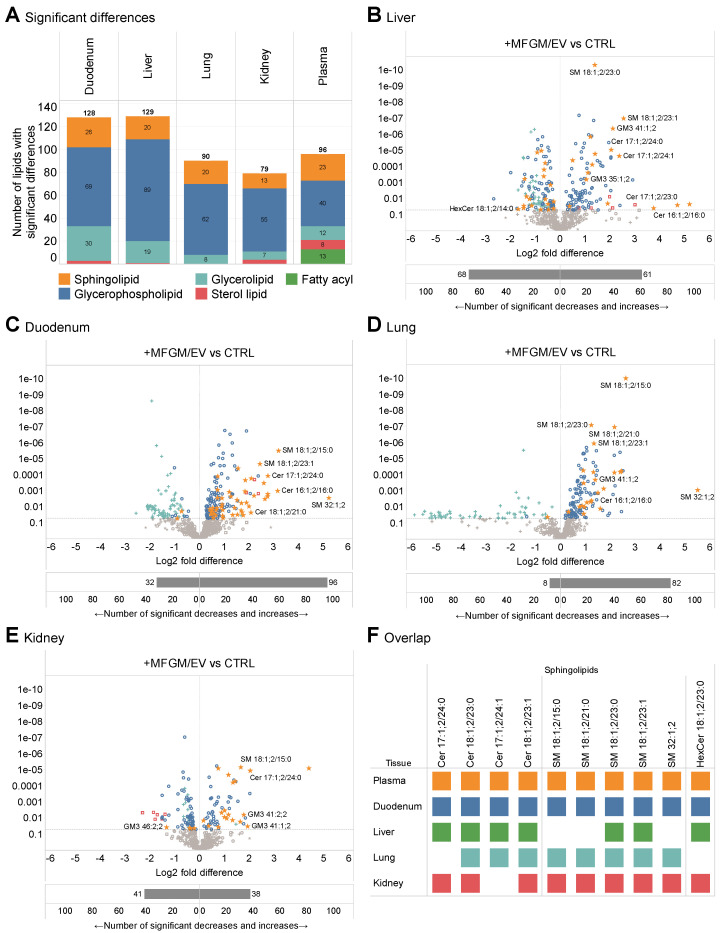
The MFGM/EV concentrate remodels the sphingolipidome of tissues. (**A**) Number of lipid molecules with a significant difference between the two dietary groups. Statistical analysis was performed using ANOVA followed by post hoc testing with multiple hypothesis correction. Lipids with q-value < 0.01 are counted as significantly different. (**B**–**E**) Volcano plots comparing lipids with significant difference between the groups of rats in the liver, duodenum, lungs, and kidneys, respectively. The horizontal line indicates the significance level at q-value = 0.05. Color-symbol scheme is the same as in Figure 2B. Underlying bar graphs show the number of significantly reduced and increased lipid abundances. Statistical analysis was performed as specified for panel A. (**F**) Overlap between significantly and consistently elevated sphingolipids in plasma and tissues of aged rats fed the MFGM/EV concentrate. Sphingolipids with q-value < 0.05 and fold difference > 1 are considered significantly different.

**Figure 4 nutrients-17-02529-f004:**
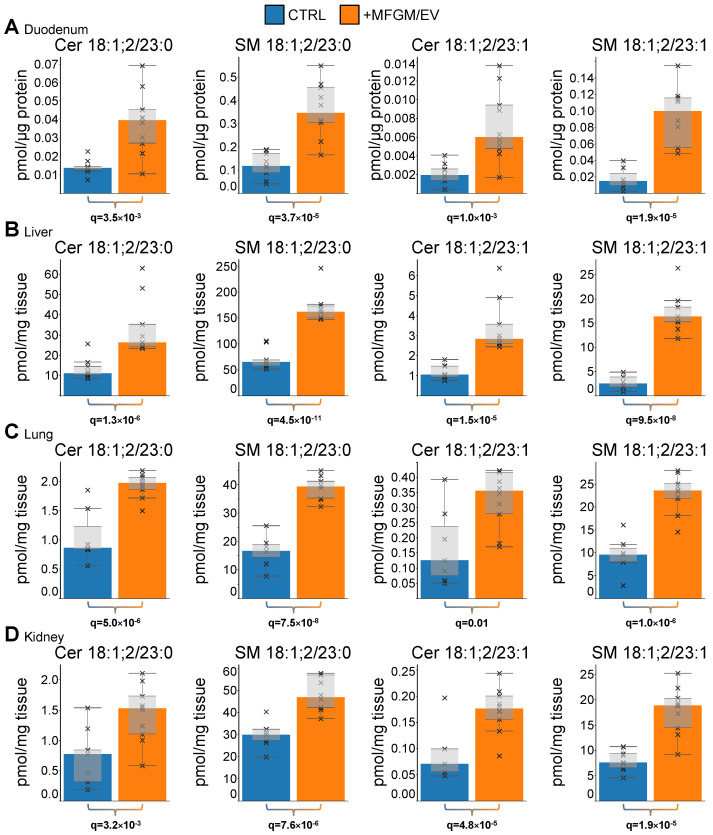
MFGM/EV supplementation promotes accretion of very-long-odd-chain sphingolipids at the whole-body level. (**A**–**D**) Levels of 23:0- and 23:1-containing sphingolipids that are significantly and consistently elevated in the indicated tissues of aged rats fed the MFGM/EV ingredient in the duodenum, liver, lungs, and kidneys, respectively. Data represent box plots, with bars indicating medians and crosses showing individual values. Statistical analysis was performed by ANOVA with multiple hypothesis correction. Significant differences are highlighted in bold.

**Figure 5 nutrients-17-02529-f005:**
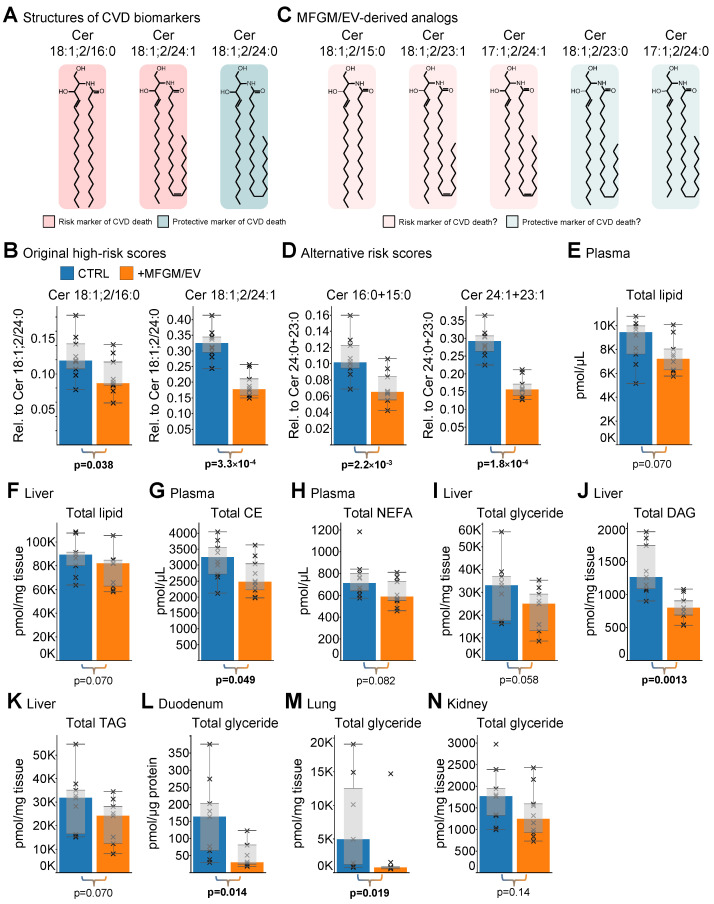
The MFGM/EV concentrate reduces lipid biomarkers associated with cardiometabolic disease. (**A**) Molecular structures of plasma ceramide biomarkers predictive of CVD mortality. (**B**) Original ceramide high-risk scores predictive of CVD death. (**C**) Molecular structures of ceramide biomarker analogs identified in aged MFGM/EV-fed rats. (**D**) Alternative ceramide risk scores that consider levels of original ceramide biomarkers and their structural analogs. (**E**–**N**) Total levels of indicated lipid categories and classes in specified biopsies. Data are represented by medians, box plots, and crosses show individual values (*n* = 7–10 rats/group). Statistical analysis was performed by a non-parametric unpaired two-sample Wilcoxon test. Significant differences (*p* < 0.05) are highlighted in bold.

**Figure 6 nutrients-17-02529-f006:**
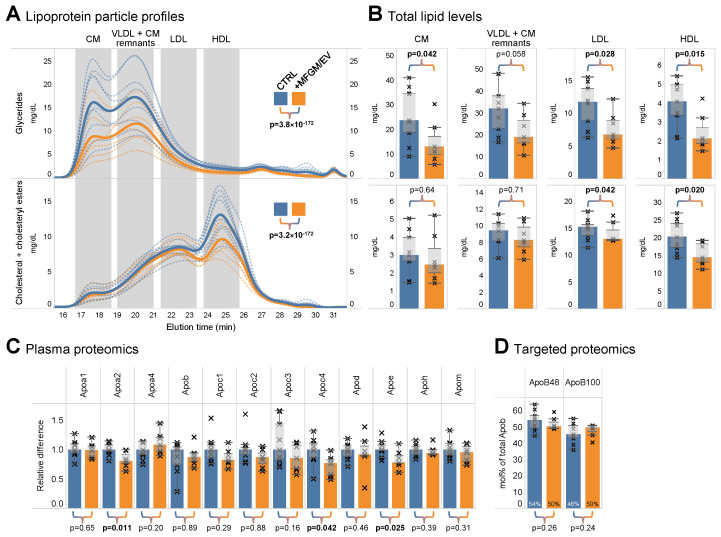
MFGM/EV supplementation modulates the profile of lipoprotein particles. (**A**) Profiles of lipoprotein particles determined by gel permeation chromatography with dual monitoring of total glycerides as well as cholesterol lipid levels. Dashed lines show individual elution profiles, and thick lines show their median profile per intervention group. Statistical analysis was performed by repeated-measures analysis using a linear mixed-effect model. Significant differences are highlighted in bold (*n* = 7–9 rats). (**B**) Total level of glycerides (upper panel) and cholesterol lipids (lower panel) in major classes of lipoprotein particles. (**C**) Profile of apolipoproteins in plasma. (**D**) Molar fractions of ApoB isoforms. The data are represented by box plots, with bars indicating medians and crosses showing individual measurements (*n* = 7–9 rats). Statistical analysis was performed by Student’s *t*-test. Significant differences (*p* < 0.05) are highlighted in bold.

**Figure 7 nutrients-17-02529-f007:**
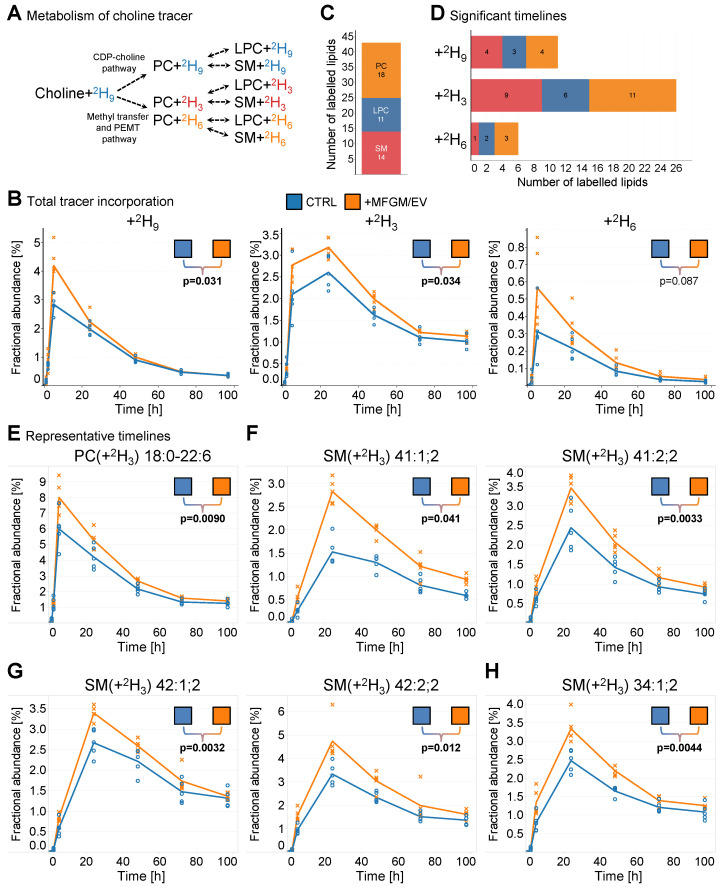
Intake of the MFGM/EV concentrate increases lipid metabolic activity. (**A**) Overview of the metabolic fates of the ^2^H_9_-labeled choline tracer. (**B**) Fractional abundances of all choline-containing lipids in plasma. Lines represent means, and crosses and circles show individual measurements (*n* = 3–5 rats/timepoint/dietary group). (**C**) Total number of lipid molecular timelines with significant differences between the dietary groups. Statistical analysis was performed by repeated-measures ANOVA and post hoc t-tests. Timelines with ANOVA F-test and two or more *t*-test *p*-values < 0.05 are considered significant. (**D**) Number of lipid molecular timelines showing significant differences between the indicated dietary groups and per label (i.e., ^2^H_9_, ^2^H_6_, and ^2^H_3_). (**E**–**H**) Representative timelines for indicated PC and SM molecules. Statistical analyses in (**B**,**E**–**H**) were performed by repeated-measures ANOVA F-test. Significant timelines are highlighted in bold (*p* < 0.05).

## Data Availability

The raw data can be found at the GNPS/massIVE repository with accession numbers MSV000097591 (plasma), MSV000097593 (liver), MSV000097595 (duodenum), MSV000097597 (kidneys), and MSV000097596 (lungs). Any additional information is available upon request.

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
