# Peer review of "Dietary Intake of a Milk Sphingolipid-Rich MFGM/EV Concentrate Ameliorates Age-Related Metabolic Dysfunction"

_nutrients, 2025, doi:10.3390/nu17152529_

Round 1

Reviewer 1 Report

Comments and Suggestions for Authors

This is an interesting research article with adequate novelty. However, some points should be addressed.

  • The conclusion section of the Abstract needs a bit more analysis.
  • The 2nd part of the 1st paragraph of the Introduction sectio (lines 37-44) needs some relevant references.
  • The sentence in lines 75-79 is too long and it should be split into two smaller sentences.
  • Before the aim of the study (Introduction) the authors should emphasize the novelty of their study, reporting the literature gap that theirs study will cover.
  • The Results and Discusion section needs more analysis concerning the Discussion part of this section.
  • The Conclusion section is too long and it should be reduce to one paragraph. Several points the Conclusion section should move to th Results and Discussion section.

Author Response

Please see attached response.

Reviewer 2 Report

Comments and Suggestions for Authors
  • Describe the sex of the rats. 
  • How were the non-survival rats managed? Were all the data discarded? 
  • Some sentences in the results repeat methodological concepts and try to eliminate repetition. 
  • How were the 10 rats from each group chosen for sampling of blood plasma and tissues for analysis of steady-state lipid and metabolite levels, as well as lipoprotein particles? 
  • Line 249 describes a group of sedentary rats not previously described in methods and were described only in Figure 1. 
  • The results and discussion sections should be separated into two different sections. The discussion must be rewritten, and the results should be compared with more papers. 
  • The first three paragraphs of the conclusions are part of the discussion. 

Reviewer 3 Report

Comments and Suggestions for Authors

This is an interesting paper and indicates the possible beneficial effects of consuming membrane material from milk and milk products. The authors have produced an impressive paper which will be of interest to a lot of people.

There are two major issues which I find unsatisfactory and require the authors’ attention. One is the lack of information on the supplements used and the other is the practice of using terms such as ‘marginally lower” and “subtly lower” to describe differences which are not statistically significant according to the criterion set by the authors.

The paper reports the feeding of a so-called “milk sphingolipid-rich MFGM/EV concentrate”. Unfortunately, the reader is given no information about this concentrate is or its contents. Section 2.1 Preparation of Supplements is devoid of essential information to allow a reader to assess this paper.

 In a previous paper by the authors (Sprenger et al 2024) (reference 18), mice were fed a “MFGM/EV-rich concentrate”. The Materials and Methods section in that paper states “Cow milk-derived MFGM/EV-rich whey concentrate, Arla® MIPRODAN 40 casein concentrate and Lacprodan® SP-9224 whey protein isolate were procured from Arla Food Ingredients (Viby J, Denmark). Information about the macronutrient composition and the molecular lipid profile of the MFGM/EV-rich whey concentrate is provided in a recent report by Sprenger et al., (2023)”. Unfortunately, the Sprenger et al. (2023) paper (which is actually Sprenger et al. (2024) Lipidomic Characterization of Whey Concentrates Rich in Milk Fat Globule Membranes and Extracellular Vesicles. Biomolecules 14 Article 55), gives the lipid profile for Lacprodan MFGM-10 (termed sweet whey concentrate) provided by Arla Foods and for two other concentrates, acid whey and buttermilk whey. Which concentrate was used in the current paper?   

One reason why it is important for the reader to know what was fed to the rats is to know the amount of fat in total and the percentage of sphingolipids in the diet.    According to the authors in their Biomolecules paper, Lacprodan MFGM-10 (sweet whey concentrate), contains 22% fat of which 35% is phospholipid.  Sphingomyelin accounts for less than 25% of the phospholipid, i.e, less than 2% of the total concentrate, and ceramides account for about one third of the sphingolipids. Therefore, the total sphingolipids in the “milk sphingolipid-rich MFGM/EV concentrate” are less than 3%.  It begs the following questions: 1. Why was this level of supplementation chosen; 2. Would the effects observed be dose-dependent; and 3. What would be the optimal level of supplementation to observe a beneficial outcome.

Line 96 refers to “the caseinate”. In their 2024 paper, the authors used Arla® MIPRODAN 40 casein concentrate which is actually calcium caseinate; was this used for the current paper? Was the caseinate mixed with Lacprodan® SP-9224 whey protein isolate as used in the authors’ previous paper?

A proper control should have included the same amount of fat as the trial diet without sphingolipids, if the effect of feeding sphingolipids was being assessed. 

It is claimed (lines 264-267) that “Altogether, these data show that the MFGM/EV supplementation is well-tolerated by the aged rats and suggests that the concentrate might have health-promoting effects, given the slightly greater weight loss in response to physical activity in the treatment group”. The authors should reconsider this statement given that the weight loss was not significantly different from that of the control group (According to Figure 1C, p = 0.077 whereas significance was set at 0.05 [line 237]).

Similarly, several statements in the first paragraph of section 3.5 need to be reconsidered.  For example, in lines 451-452, it is stated that “the supplement had a general lipid lowering effect”; however, in the total lipid data in Figure S4, there was no significant difference in any of the analyses. In subsequent lines, the authors state “the supplementation led to a 1.3-fold (p=0.070) and 1.1-fold (p=0.070) lower total lipid concentration in plasma and liver, respectively; these are non-significant differences and should not be depicted as lower levels due to supplement feeding. Other statements in the same paragraph (lines 455, 457, 459) refer to “marginal” and “subtle” lowering; such terms have so place in scientific reporting. These differences are non-significant according to the criterion set in line 237.

The conclusions are well-stated although some parts probably belong better in Discussion than Conclusions.  I was pleased to see the authors address the issue of the mechanism(s) by which dietary sphingolipids directly affect the composition of blood and tissue lipids (lines 592-591).  Furthermore, they rightly raise the issue of lack of understanding of the mechanisms that govern the health-promoting effects of milk sphingolipids (lines 603-621). I look forward to seeing future research results which address these mechanisms.

Reviewer 4 Report

Comments and Suggestions for Authors

Dietary intake of a milk sphingolipid-rich MFGM/EV concentrate ameliorates age-related metabolic dysfunction

This study investigates the effects of a milk sphingolipid-rich MFGM/EV concentrate on age-related metabolic dysfunction in aged sedentary rats. The manuscript  evidences that MFGM/EV supplementation modulates sphingolipid metabolism, reduces cardiometabolic risk markers, and improves lipoprotein turnover. The findings are novel and clinically relevant, particularly for elderly populations. However, some aspects require clarification or further discussion to strengthen the manuscript.

  1. Rats in the control group were supplemented with a drink containing a milk lipid-deficient whey protein-caseinate blend, but the matching of calories and nutritional composition between it and the experimental group was not explained. It is recommended to supplement relevant information to eliminate the potential impact of calorie differences on the results.
  2. The study only used male rats and did not take into account the impact of gender differences on the results. It is suggested to discuss the potential impact of gender restrictions on the research, or to supplement the experimental data of female animals.
  3. Section 2.2 of the Methods should specify the rationale for reducing the supplement volume mid-study (300 mL → 200 mL). Was this due to palatability or supply limitations?  
  4. The results section presents the treatment group and control group, water maze test, sedentary rats and adolescent mice. However, the specific experimental groups and operation processes are not introduced. It is suggested that more detailed information be provided in the methods section to help readers understand the article more clearly.
  5. Line 270: the Morris water maze test was mentioned but no further discussion was given. As the cognitive results will be published separately, it is recommended to briefly summarize the key findings to provide background.
  6. The study cited supplementary tables and charts, whilecontents were not fully discussed in the study. It is suggested that the results of the supplementary materials be cited and explained in more detail in the article.
  7. The baseline metabolic status of the rats (such as whether they were obese or insulin resistant) was not discussed in the study. It is recommended to supplement baseline data to eliminate the interference of individual differences on the results.
  8. The study did not evaluate the long-term safety of MFGM/EV supplements (such as liver function,inflammatory markers or Usage safety threshold). It is suggested to supplement relevant data to support the safety of its clinical application.
  9. The clinical significance of some lipid biomarkers in the study (such asCer 18:1;2/23:1) has not yet been introduced in humans. It is suggested to discuss the potential application value of these markers in human research.
  10. The following are the spelling mistakes and grammar issues found in the article, along with the corresponding suggestions for improvement:
    • Line 242: "regiments" should be"regimens."
    • Page 7, Figure 1 Caption"Repeated measures ANOVA F-test shows no sig-nificant differences". Suggestion: "sig-nificant" is a line break error and should be merged into "significant".
    • Page 12, Results: "the heath-promoting effects",it is suggested that "heath"should be "health".

This work has high potential to advance nutraceutical strategies for age-related metabolic disorders.

Round 2

Reviewer 1 Report

Comments and Suggestions for Authors

The authors have significantly improved their manuscript.

Reviewer 2 Report

Comments and Suggestions for Authors

Conclusions must be shortened.

Results and discussion should be separated